# New Self-Healing Metallosupramolecular Copolymers with a Complex of Cobalt Acrylate and 4′-Phenyl-2,2′:6′,2″-terpyridine

**DOI:** 10.3390/polym15061472

**Published:** 2023-03-16

**Authors:** Evgeny S. Sorin, Rose K. Baimuratova, Igor E. Uflyand, Evgeniya O. Perepelitsina, Denis V. Anokhin, Dmitry A. Ivanov, Gulzhian I. Dzhardimalieva

**Affiliations:** 1Federal Research Centre of Problems of Chemical Physics and Medicinal Chemistry RAS, 142432 Chernogolovka, Russia; sorin_es@mail.ru (E.S.S.); rozab@icp.ac.ru (R.K.B.);; 2Department of Fundamental Physical and Chemical Engineering, Lomonosov Moscow State University, 119234 Moscow, Russia; 3Moscow Aviation Institute, National Research University, 125993 Moscow, Russia; 4Department of Chemistry, Southern Federal University, 344090 Rostov-on-Don, Russia; 5Institut de Sciences des Matériaux de Mulhouse (CNRS UMR 7361), 68057 Mulhouse, France

**Keywords:** self-healing, metal-containing monomers, terpyridine-containing polymers, coordination polymers

## Abstract

Currently, the chemistry of self-healing polymers is aimed not only at obtaining materials with high self-healing efficiency, but also at improving their mechanical performance. This paper reports on a successful attempt to obtain self-healing copolymers films of acrylic acid, acrylamide and a new metal-containing complex of cobalt acrylate with a 4′-phenyl-2,2′:6′,2″-terpyridine ligand. Samples of the formed copolymer films were characterized by ATR/FT-IR and UV-vis spectroscopy, elemental analysis, DSC and TGA, SAXS, WAXS and XRD studies. The incorporation of the metal-containing complex directly into the polymer chain results in an excellent tensile strength (122 MPa) and modulus of elasticity (4.3 GPa) of the obtained films. The resulting copolymers demonstrated self-healing properties both at acidic pH (assisted by HCl healing) with effective preservation of mechanical properties, and autonomously in a humid atmosphere at room temperature without the use of initiators. At the same time, with a decrease in the content of acrylamide, a decrease in the reducing properties was observed, possibly due to an insufficient amount of amide groups to form hydrogen bonds through the interface with terminal carboxyl groups, as well as a decrease in the stability of complexes in samples with a high content of acrylic acid.

## 1. Introduction

To date, the use of self-healing polymers capable of fully or partially restoring their original properties and/or functions makes it possible to solve many problems associated with the wear of polymer products under the influence of mechanical load or the environment [1,2,3]. In terms of the need for initiation, healing processes can be divided into autonomous and non-autonomous (requiring exposure to temperature or radiation) [4]. Healing can also be classified by nature into external and internal. External requires the encapsulation of ‘healing’ agents that, when damaged, fill the formed microfracture [5,6]. Internal is caused by the presence of different types of reversible–destructive bonds (disulfide bonds [7] and Diels–Alder reactions [8] for covalent–reversible interaction; hydrogen bonds [9], π-π stacking [10] and cation-π interaction [11], ionic [12], host–guest [13] and metal–ligand (M–L) interactions [14] for non-covalent reversible interactions).

It is a class of internal healing polymers that has been actively developed over the past decade [15,16], since such materials will better retain their original functionality during recovery after damage, and the development of such materials is more convenient from a technological point of view. However, now it is quite difficult to create autonomous systems with an internal recovery mechanism. In addition, practical application requires imparting high mechanical properties to these materials.

One possible solution to this problem is the use of metallopolymers due to the inclusion of reversible non-covalent M–L interactions [17]. By varying the metal ion or pH of the system, it is possible to regulate the bonding strength and binding dynamics of the system, which makes it possible to create polymers with adjustable mechanical properties [18,19,20,21].

Interest in the functionalization of self-healing polymers has also increased in recent years. Metallopolymers can be useful in this aspect, since the presence of flexible M–L bonds and the ability to change both the metal ion and the ligand make this class of polymers the most promising for creating materials with a wide range of practical applications, such as luminescent [22], photo- [23], thermosensitive [24] and electrically conductive polymers [25]. In this regard, interesting monomers are terpyridine-containing systems, especially those based on the so-called Cronke motif which have a functionalized phenyl fragment in the 4′-position [26,27,28]. A variety of coordination chemistry, affinity for binding different ions and redox properties make it possible to create a large variety of metallosupramolecular architectures based on phenylterpyridine complexes with a wide range of applications, such as luminescent and electrochemical sensors, OLEDs, etc. [29,30].

In most works published over the past 5 years [31,32,33,34,35,36], such metallopolymer systems were obtained as follows: polymers containing chelating ligands with donor fragments such as pyridine and its derivatives at the terminal sites were impregnated with salts of various transition metals. However, it has never been possible to obtain a material that simultaneously has high strength characteristics and autonomous internal healing: polymer healing occurs only in the presence of an external influence, such as temperature or light, which makes it difficult to use such systems in practice.

In this work, we propose a one-step method for the preparation of new types of self-healing copolymers of acrylic acid (AAc), acrylamide (AAm) and a mixed ligand metal-containing monomer (MCM).

It is known that polyacrylamide (polyAAm) is the main polymer used in creating self-repair hydrogels [37]. The main disadvantage is extreme extensibility and compressibility after swelling in aqueous solutions [38]. To overcome the above disadvantage, AAm is usually copolymerized and crosslinked with organic crosslinkers to form interpenetrating networks [39]. For example, copolymerization with acrylic acid creates physical crosslinks in the polymer, with different strengths between randomly distributed repetitive units AAm and AAc [40]. There are known methods of increasing the strength and extensibility of the network by molecular complexation with metal ions; in such systems, coordination bonds serve as permanent crosslinking [38,41,42].

Previously, it was shown that the inclusion of Zn(II) and Cu(II) metal-chelate monomers into the poly(AAc-AAm) copolymer affects the reactivity of the double bond during polymerization [43]. This work also demonstrated the ability of such systems to achieve high efficiency of self-healing of mechanical properties, which is most likely associated with the presence of an additional system of double bonds in MCM. In this regard, it was of interest to expand the range of MCMs based on 4′-phenyl-2,2′:6′,2″-terpyridine (PhTpy). Previously, we have obtained and characterized a mixed-ligand cobalt acrylate phenylterpyridine complex (CoAcr_2_PhTpy) [44]. It should be noted that the synthesized CoAcr_2_PhTpy has a high effective heat of double bond opening (116.8 J/g) and crystallinity. The present work is devoted to the synthesis and study of new self-healing metallopolymers based on AAc, AAm and a metal-containing cobalt acrylate complex with a phenylterpyridine ligand. The incorporation of MCM directly into the polymer chain not only improves the mechanical properties of the final polymer films, but also leads to an almost unique effect for such systems at present—internal autonomous healing.

## 2. Materials and Methods

### 2.1. Starting Materials

Acrylamide (AAm, >99%) and potassium persulfate (KPS, ≥99.0%) purchased from Aldrich (Moscow, Russia) were used without additional purification. Acrylic acid (AAc stabilized with hydroquinone monomethyl ether for synthesis, >98%) purchased from Sigma-Aldrich (Moscow, Russia) was used after additional purification by vacuum distillation. The preparation, physical and chemical properties of a cobalt-containing monomer based on AAc and PhTpy were described earlier (Figure 1) [44].

### 2.2. Synthesis of Polymers and Film Formation

The polymers were synthesized by free-radical polymerization of an aqueous solution of the precursor at a certain ratio of AAc, AAm and a third comonomer with the PhTpy ligand. Polymerization was carried out in polypropylene ampoules with a tight screw cap. When the initiator (K_2_S_2_O_8_) was added, the reaction mixture instantly polymerized in the frontal mode at room temperature. The composition of precursor solutions is given in Table 1. For convenience, the table shows the initial weight of AAm and AAc, since their molar weight is almost the same.

As a result of polymerization, dense light orange hydrogels were obtained. The films were formed from an aqueous polymer solution in special open glass molds by air drying at room temperature. Drying was carried out to constant mass values. Once the film had completely detached from the mold, the film was inverted to achieve maximum drying on both sides.

### 2.3. Analytical Methods

#### 2.3.1. Analysis of Structure and Thermal Properties

C, N, H content was studied on a Vario Micro cube elemental analyzer (Elementar GmbH, Hanau, Germany), and the content of cobalt was studied on an AAS-3 atomic absorption spectrometer (Zeiss, Jena, Germany).

pH was measured by portable a pH-meter pH-410 (Akvilon, Saint Petersburg, Russia) with a glass combination pH electrode (reference electrode—AgCl).

IR analysis was performed on a Bruker ALPHA Fourier-IR spectrometer (Bruker Optik GmbH, Ettlingen, Germany) equipped with a single reflection diamond prism; the penetration depth for a medium with a sufficiently deep refractive index (2.43) at 1000 cm^−1^ is 1.66 microns. UV spectrometry was carried out on a SPECS-SSP-705-1 spectrometer (JSC Spectroscopic Systems, Moscow, Russia).

Analysis of the molecular weight distribution was carried out on a «Waters» liquid chromatograph equipped with a «Waters2414» differential refractometric detector and a «PDA 996» diode array spectrophotometric detector (Waters Corporation, Milford, NJ, USA). A PLgel 5 μm MIXED-C column was used. The eluent used was N-methylpyrrolidone (NMP) + LiCl (1.0 g LiCl/0.5 L NMP), elution rate 1 mL/min, T_col_ = 70 °C, T_ref_ = 50 °C. Standard polystyrene samples with MM from 580 to 3.7106 Da were used to construct the calibration curve. The obtained chromatograms were processed using the «Empower» software. The polymer samples were dissolved in NMP+LiCl (concentration 20 ÷ 40 mg/mL) and the solution was filtered through a 0.2 µm PTFE filter Anatop25 (“Whatman”).

The structural investigation was carried out at the BM26 beamline of the European Synchrotron Radiation Facility (ESRF) in Grenoble, France. The beamline is equipped with Pilatus 1M (SAXS, q-range 0.009–0.46 Å^−1^) and Pilatus 300k (WAXS, q-range 0.65–5.2 Å^−1^) detectors. The measurements were made in the transmission geometry using X-ray photons with an energy of 13 keV (wavelength 0.954 Å), and the spot size of the beam on the sample was 0.65 mm × 0.65 mm. The two-dimensional diffraction patterns were analyzed using a library developed in the Igor Pro Program package (Wavemetrics Ltd., Portland, OR, USA) [45,46].

XRD studies were performed on a PANalytical Aeris Research diffractometer (Malvern PANalytical B.V., The Netherlands) with CuK_α_ radiation (λ_Cu_ = 1.5406 Å). Electron microscopic studies of the obtained polymer films were carried out on a Zeiss LEO SUPRA 25 scanning auto-emission electron microscope (Carl Zeiss, Jena, Germany) combined with an EDX system for microanalysis.

Surface topology films were observed with a confocal laser scanning microscope (Optelics Hybrid, Lasertec Corp., Tokyo, Japan) equipped with a 5× objective.

Differential scanning calorimetry (DSC) and thermogravimetric analysis (TGA) curves were recorded on a METTLER TOLEDO DSC822e differential scanning calorimeter (Mettler Toledo, Greisensee, Switzerland) and a METTLER TGA/SDTA851e thermogravimetric analyzer (Mettler Toledo, Switzerland), conjugated with quadrupole mass spectrometer QMS 403C Aeolos and METZSCH STA 409 PC/PG. The samples were heated in a nitrogen atmosphere at a heating rate of 10 K∙min^−1^ and 20 K∙min^−1^.

#### 2.3.2. Physical and Mechanical Tests and Assessment of Self-Healing Efficiency

Physical and mechanical tests were carried out on a universal tensile machine Zwick/Roel TC-FR010 (ZwickRoell GmbH & Co. KG, Ulm, Germany) at room temperature at a tensile speed of 1 mm/min. For the study, two-sided blade-shaped samples of length L = 75 mm, tear-off width d = 4 mm and an average thickness of 1 mm were cut from tear-off width (Russian GOST 270-75 type III) films. All series of samples were presented from more than 5 blades, and the discrepancy between the test results did not exceed 10%.

After the test, the two parts of the ruptured sample were brought into contact and the rupture site was moistened with 0.1 M HCl to accelerate the healing process.

Healing efficiency was evaluated by comparing the tensile strength and elastic moduli of the healed specimen with the original specimen according to Equations (1) and (2):(1)$$Z_{\mathrm{stress}}=\frac{\sigma_{h}}{\sigma_{0}}\times 100\%,$$
(2)$$Z_{E}=\frac{E_{h}}{E_{0}}\times 100\%,$$
where σ_h_ and E_h_ are the tensile strength and modulus of elasticity of the healed sample and σ_0_ and E_0_ are the tensile strength and modulus of elasticity of the original sample.

## 3. Results and Discussion

### 3.1. Structure and Composition of Copolymers

#### 3.1.1. Elemental Analysis

The composition of the obtained copolymers (Figure 2) was determined by elemental analysis (Table 2).

According to the results of elemental analysis, the samples with the initial equimolar ratio and the highest AAc content contained ~11% water of the residual weight, while the sample with the highest AAm content contained 23% water. Most likely, this is due to the ability of AAm units to form strong hydrogen bonds with solvent molecules due to the presence of the most developed spatial network of hydrogen bonds.

#### 3.1.2. IR- and UV-Vis-Spectrometry

Samples of the formed copolymer films were analyzed spectroscopically by the method of attenuated total reflectance (ATR) in the wavenumber range from 4000 cm^−1^ to 360 cm^−1^ (Figure 1).

In the ATR spectra of a copolymer sample with a high content of AAm (Figure 1, copolymer one) at 3500–3150 cm^−1^, wide absorption intensity regions are observed, indicating the presence of intra- and intermolecular hydrogen bonds in the polymer associated with stretching vibrations of the -NH bond of the AAm chain and the stretching vibrations of the -OH groups of AAc units. The absorption bands at 2959 and 2875 cm^−1^ can be attributed to asymmetric and symmetric stretching vibrations of the C-H group of the main chain of the ternary copolymer [47]. The shift of the C=O valence band to 1649 cm^−1^ indicates the formation of hydrogen bonds between -COOH and -CONH_2_ [48]. The absorption in the region of 1450–1414 cm^−1^ can be attributed to the relaxation of the asymmetric stretching of the CoO-C bond in the carboxylate units of the polymer. Intense absorption bands at 1022 and 1058 cm^−1^ in samples with a higher content of AAc (copolymer two and copolymer three) can be attributed to vibrations of the C-O-C bond of the anhydride group CO-O-CO, which apparently formed due to the intramolecular cyclization observed because of local overheating during the frontal polymerization of the terpolymer. As the vibration intensity of the carbonyl group decreased, an intense signal appeared at 1256 cm^−1^, which, as a rule, corresponds to the C–N bond in polycyclic amines.

According to the UV-vis absorption spectra of the copolymers (Figure 2), an increase in the AAm content leads to an increase in the metal-ligand charge transfer (MLCT) signal observed in the region from 430 to 500 nm, indicating the successful incorporation of the metal complex into the polymer chain. A similar effect can also be observed in the IR spectra: absorption in the region of 550–500 cm^−1^, related to the Co-O vibration, symbiotically depends on the AAm content in the system. Such a dependence can most likely be explained by the different stability of the complex at various monomer ratios, which, among other things, may additionally affect the healing properties.

#### 3.1.3. X-ray Studies

There are no peaks on the SAXS curves (Figure 3A), i.e., there is no supramolecular organization. The WAXS curves (Figure 3B) show a broad peak q = 1.7 Å^−1^ corresponding to a distance of 3.7 Å. This is the characteristic thickness of the benzene ring, so this peak can be attributed to the formation of small aggregates of terpyridine ligands. For the sample of copolymer one, the most ordered structure was observed; there was also a peak q = 2.7 Å^−1^ (d = 2.3 Å), which can be related to the distance between the cobalt atoms. Thus, the sample with the highest AAm content with the packing of ligands and metal centers of the complex was the most ordered, which may be due to the above-mentioned higher stability of the complex at this ratio of monomers (see Figure 1 and Figure 2). The copolymer with an excess of AAc only had ligand packing, while there was no ordering for the sample with an equimolar ratio.

High-angle diffraction patterns of the obtained samples (Appendix A) also show the highest ordering of copolymer one, which is consistent with the results of the SAXS and WAXS studies. All copolymers had a peak in the 22° region, corresponding to an interplanar distance of 4 Å = 0.4 nm, which is approximately equal to half of the length of the benzene ring [49].

The formation of crystallinity regions of such copolymers can be due to hydrogen bonds. Interestingly, the structure of the thermolysis product of copolymer one was indirectly preserved, as seen in the SEM images (Appendix A).

#### 3.1.4. Thermal Properties of Polymers

According to the TGA and DTG curves (Table 3, Figure 4), the obtained copolymers decomposed in four main stages. The first stage proceeded monotonically up to 100–130 °C, with a weight loss of no more than 2–3 wt.% and was accompanied by a significant endothermic effect on DSC (Figure 5). This is due to the loss of physically adsorbed water molecules (*m*/*z* = 18; 17) according to ion-fragmentation mass spectrometry. Weight loss up to 15 wt.% was observed in the second stage from 150 to 240 °C, with maximum decomposition rates from 199 to 229 °C. The third stage of decomposition proceeded at a temperature of 230 to 305 °C, with a maximum decomposition rate at 280 to 291 °C and a total weight loss of up to 25–30 wt.%. The fourth main decomposition stage, observed at temperatures above 310 °C, was associated with decarboxylation (*m*/*z* = 28; 44) and decomposition of the polymer backbone into AAm fragments, and was accompanied by a total weight loss of up to 30–35 wt.%.

It should be noted that an increase in the AAm content of more than an equimolar value in the composition of the copolymer slightly changes the nature of decomposition, as can be seen in Figure 4b (Copolymer one). This effect is probably associated with the ability of AAm units to strongly interact with solvent molecules due to hydrogen bonds, which correlates with the previously obtained data of elemental analysis (see Table 2) and IR spectra (see Figure 1).

DSC curves show two pronounced endothermic effects, the first of which corresponded to a partial removal of the solvent from the polymer films; the second effect was observed in the range of maximum mass loss rates according to DTG curves, which may indicate polymer decomposition. Glass transition temperatures in the studied range could not be detected.

### 3.2. Mechanical and Self-Healing Properties of Metallopolymers

The healing efficiency was evaluated by comparing the values of the maximum strength and moduli of elasticity of the healed sample (σ_h_, E_h_) with the original (σ_o_, E_0_) according to Equation (1). The healing process was accelerated with a 0.1 M HCl solution (Figure 6), since at low pH values the terminal carboxyl groups are completely protonated, which increases the hydrogen bond interaction [50].

The tests were carried out on a series of at least five specimens for each copolymer. The discrepancy between the experiments did not exceed 10%. The stress–strain (σ-ε) curves for the original and healed samples are shown in Figure 7.

According to the results of mechanical tests, the sample with the highest AAm content (copolymer one) had the best strength characteristics and resisted stress of 125 MPa, and had a modulus of elasticity of 4.3 GPa, which is almost a record for such systems today. From the point of view of resistance to critical stresses and the possibility of practical application, the healing efficiency should be primarily evaluated by comparing the tensile strength values of the original and healed sample, since the strength in this case will be determined by the crosslinking of the sample, and the elastic modulus by the kinetic flexibility of the chains [51]. A sample with an equimolar ratio of initial monomers has the best healing efficiency in terms of strength recovery—62%. The calculated mechanical parameters and healing efficiencies (Equations (1) and (2)) are presented in Table 4 and Table 5.

Healing, similarly accelerated by the addition of 0.1 M HCl, is shown in Figure 8, Figure 9 and Figure 10. A similar recovery process was observed for all specimens: the through-cut was tightened, leaving a semblance of a “scar” in its place, and surface small cracks and cuts were almost completely healed. Interestingly, the metallopolymer sample with an excess of AAc, which showed worse mechanical results and low recovery efficiency, was also inferior in terms of healing time, although at this stage this assessment is mostly qualitative.

A completely autonomous internal healing in air in a humid environment was also found; the crack healed for three days for all copolymers (Figure 11).

For the initial analysis of HCl-accelerated healing, a model reaction of a metal complex with hydrochloric acid was considered. The UV-visible spectra of the metal complex before and after the addition of hydrochloric acid are shown in Figure 12.

In the spectra of the UV-visible region after the addition of HCl, an initial decrease in absorption in the region of 400–530 nm was observed, which may indicate partial or complete decomposition of the complex upon interaction with the acid. In this case, the formation of cobalt bis-terpyridine complexes known in the literature for such systems [52,53] is quite probable, as evidenced by the change in the color of the solution after the addition of HCl (Figure 3).

By itself, such a reaction in the polymer (Figure 2) can already contribute to the healing of the material. On the one hand, the ongoing termination of the Co-O bond increases the kinetic flexibility of the polymer chain, which facilitates the healing process. This is consistent with healing efficiency in relation to the modulus of elasticity (Table 5). The healing efficiency increases symbiotically with the AAm content in the system, which positively affects the stability of the complex in the system (see Figure 1 and Figure 2). On the other hand, this interaction can be an additional source of reversible M–L bonds on opposite sides of cracks/microcracks during damage.

IR spectra of the healed copolymers were obtained to evaluate the HCl-accelerated healing processes (Figure 13).

The IR spectra of the copolymers after healing show signals at 550–500 cm^−1^ associated with Co-O oscillation. The rearrangement of the complex in the polymer upon the addition of HCl (Figure 3 or a similar reaction) probably proceeded less actively due to the high viscosity of the system, because of which not all chains were terminated in the wetting region.

A significant increase in the signal range at 3150–3050 cm^−1^ was observed for a copolymer with an equimolar ratio, indicating intra- and intermolecular hydrogen bonding interactions, and at 1410 cm^−1^, which is associated with the relaxation of asymmetric stretching of the CoO-C bond in the carboxylate units of the polymer. The first fact can possibly be attributed to the partial hydrolysis of AAm units in an acidified medium. The asymmetric stretching of the CoO-C bonds was probably caused by the stacking interaction of the benzene rings when the polymer was rearranged during the healing process.

Thus, the general mechanism of accelerated healing of polymers with the help of HCl is a combination of processes: breaking the polymer chain along the Co-O bonds, which causes an increase in kinetic flexibility and the appearance of an additional source of reversible M–L bonds, partial hydrolysis of amide groups, followed by an increase in intermolecular hydrogen interaction and π-π-stacking of benzene rings.

It is very likely that additional rearrangements of the metallocomplex in the chain also occurred, since internal healing was observed completely autonomously (Figure 11). However, additional research of the mechanism is currently required to test this assumption.

In order to evaluate the effect of introducing a metal-containing complex into the polymer system, an attempt was made to obtain acrylamide and acrylic acid copolymer films and to study their mechanical characteristics and healing efficiency. The stress–strain (σ-ε) curves for the original and healed samples without CoAcr_2_PhTpy are shown in Figure 14 and their characteristics are shown in Table 6. The sample with the highest acrylamide content could not be dissolved, so the film of this specimen could not be formed.

The strength of both samples increased relative to metallopolymers. Copolymers without the inclusion of the phenylterpyridine complex showed a significantly worse healing efficiency in terms of maximum strength (virtually no elastic modulus was retained). There was also almost no autonomous healing after three days (Figure 15).

Thus, the incorporation of CoAcr_2_PhTpy into the chain leads to a multiple enhancement of the self-healing properties.

Although the metal-containing complex should be an additional crosslinking agent, it probably reduced the molecular weight of the final copolymers, so that films of samples with the highest acrylamide content can be obtained in this way, thereby creating high-strength materials with autonomous internal healing.

The key question is also about the role of the supramolecular structure in the healing mechanism. At present, it is often stated that the self-healing properties require a supramolecular organization of the polymer, which is absent in this system (Figure 3A). The best healing efficiency in terms of tensile strength (Table 4) was demonstrated by copolymer two, which also had neither a metal center packing nor a ligand packing (Figure 3B).

From the obtained data, it can be assumed that the kinetic flexibility, which affects the magnitude of the change in the entropy of polymer chains, plays a predominant role in the creation of a new surface during healing.

Copolymer one had the best mechanical properties. This can be explained both by the presence of the most developed spatial network of hydrogen bonds and by the local packing of ligands and metal centers of the complex. The second fact may be associated with the stability of the complex noted above at a given ratio of monomers and, consequently, its higher content in the final polymer chain. However, an important question remains, which we will try to answer in a future study: why exactly the sample with an equimolar ratio of monomers (copolymer two) did not form aggregates, unlike copolymer one and copolymer three. Understanding this process can elucidate the mechanism of the healing process.

## 4. Conclusions

New self-healing metallopolymers of acrylamide, acrylic acid and a complex of cobalt acrylate with 4′-phenyl-2,2′:6′,2″-terpyridine have been obtained. The incorporation of the metal complex directly into the polymer chain allowed high mechanical properties to be achieved for this system. The initial copolymer one achieved a tensile strength of 122 MPa and a modulus of elasticity of 4.3 GPa, which is a near-record for such materials. Copolymer two had a good healing efficiency with a tensile strength of 62%, while having a relatively high strength of 50 MPa and a modulus of elasticity of 3.4 GPa. Copolymer one and copolymer two also showed a high recovery efficiency in elastic moduli of 90% and 65%, which is probably due to the increased kinetic flexibility of the polymer chain during accelerated healing with 0.1 M HCl. In addition, all samples showed a completely autonomous internal healing. All of the above makes this system promising for practical use in the future. However, many questions remain about the mechanism of self-healing and the role of supramolecular organization, the answers to which require a more detailed study of the structure of the monomer and polymer and the kinetics of their formation, as well as expanding the range of initial metal complexes with phenylterpyridine, for which further work will be aimed.

## Data Availability

The data presented in this study are available on request from the corresponding author.

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
