# Peer review of "New Self-Healing Metallosupramolecular Copolymers with a Complex of Cobalt Acrylate and 4′-Phenyl-2,2′:6′,2″-terpyridine"

_polymers, 2023, doi:10.3390/polym15061472_

Round 1

Reviewer 1 Report

This manuscript presented a highly interesting strategy to construct self-healing polymer system involving three different supramoleclar interactions: hydrogen bonding, pi-pi stacking, metal-ligand interaction. This novel copolymer system was well investigated in this manuscipt by detailed structural and property characterizations. The orgainizaion of this manusciprt is reasonable.  The referee thinks the overall quailty of this manuscipt is good enough for publishing in Polymers. The following commnets should be considered:

1. The ligand metal-containing monomer here is actually difunctional (two double bonds). Is it possible that there are a few chemically cross-linked sites in the copolymer network, not only the physically cross-linked sites? If so, it's recommened to indicate this in Scheme 2.

2. How the synthesized copolymers were dried? The drying procedure should also be described in details since there is physical trapped water in the copolymer network by hydrogen bonding. It's important to follow the same drying procedure for the three copolymers studied in this manuscript.

3. Why did not the authors further investigte the self-healing properties of copolymers healed by the autonomous internal healing in air in a humid environment, ie, the self-healing efficiency? After all, as the authors mentioned in the introduction part, the autonomous internal healing is more meaningful for partical applications.

4. How big is the difference in mechanical properties and self-healing properties between the AAc-AAm copolymer and AAc-AAm-MCM copolymer with similiar composition?

Author Response

Dear Reviewer 1,

We are grateful for time and effort that you paid for reviewing our article. All the comments have helped us to greatly improve the manuscript. The attached Word document contain point-by-point response to comments and concerns.

Sincerely, the authors of the article

Reviewer 2 Report

See the atached document

Author Response

Dear Reviewer 2,

We are grateful for time and effort that you paid for reviewing our article. All the comments have helped us to greatly improve the manuscript. The attached Word document contain point-by-point response to comments and concerns.

Sincerely, the authors of the article

Reviewer 3 Report

In the submitted manuscript by Dzhardimalieva and coworkers, three copolymers derived from acrylic acid (AAc), acrylamide (AAm) and a metal-containing monomer (MCM) – cobalt acrylate with a phenyl terpyridine ligand – are successfully prepared for the first time via free radical polymerization. The copolymers contain different contents of AAc and AAm, with less than 1% MCM incorporated. All three samples are thoroughly characterized by ATR/FT-IR and UV-vis spectroscopy, elemental analysis, DSC, TGA, SAXS, WAXS and XRD studies. In addition, these samples show self-healing behaviors at acidic condition or autonomously in a humid atmosphere. Investigation of mechanical properties before and after healing reveals the self-healing behavior is related to the contents of AAc and AAm in copolymer samples, although the tensile strength and elastic modulus cannot be fully recovered in all three samples. Furthermore, the mechanism of self-healing is briefly discussed but still needs to be elucidated in the future.

Overall, the authors have demonstrated the synthesis of new metallosupramolecular copolymers using a cobalt containing monomer and studied their self-healing behavior. This work could be suitable for Polymers, and acceptance with major revisions is recommended after addressing the points listed below.

Major points:

1)     Background about self-healing materials based on AAc and AAm copolymers should be included in the introduction.

2)     Self-healing behaviors of samples without the cobalt-containing monomer (only AAc and AAm) should also be displayed. These control experiments would be helpful to show the role of cobalt phenyl terpyridine in the self-healing process.

3)     Tensile strength and elastic modulus results as well as the healing efficiencies should also be showed and discussed in the autonomous healing experiments.

Minor points:

What is the pH during the copolymerization? It should be clarified in the synthetic procedure as this could affect the reactivity ratio of AAc and AAm.

Line 107 – The term “Aam” should be “AAm”.

Table 2 – Molecular weights and dispersties of copolymer 1 and 2 should also be included.

Figure 6, Figures 8-11 – The characterization technique to obtain these images should be included. Are these optical microscopic images?

Line 400 – “copolymer 3” should be “copolymer 1”.

Author Response

Dear Reviewer 3,

We are grateful for time and effort that you paid for reviewing our article. All the comments have helped us to greatly improve the manuscript. The attached Word document contain point-by-point response to comments and concerns.

Sincerely, the authors of the article

Round 2

Reviewer 2 Report

The authors answered to all my remarks.

Reviewer 3 Report

I'm pleased to see that the authors have addressed most of my concerns in their latest manuscript revision, and they have also provided detailed explanations for those that couldn't be resolved. Based on these revisions, I believe that the manuscript is now suitable for publication.